Does regional diversity recover after disturbance? A field experiment in constructed ponds

Woods Lauren M. 1 2
Biro Elizabeth G. 1
Yang Muxi 3
Smith Kevin G. kgsmith@davidson.edu 4
1 Department of Biology, Tyson Research Center, Washington University in St. Louis , St. Louis , MO , United States
2 Department of Biology, Davidson College , Davidson , NC , United States
3 Tyson Research Center, Washington University in St. Louis , St. Louis , MO , United States
4 Department of Biology, Department of Environmental Studies, Davidson College , Davidson , NC , United States
Sundermann Andrea
Electronic publication date: 2016 Oct 18
Publication date: 2016
Volume: 4
Electronic Location ID: e2455
Received 2016 May 17; Accepted 2016 Aug 17
Copyright: ©2016 Woods et al.
Copyright year: 2016
Copyright holder: Woods et al.
License: This is an open access article distributed under the terms of the Creative Commons Attribution License, which permits unrestricted use, distribution, reproduction and adaptation in any medium and for any purpose provided that it is properly attributed. For attribution, the original author(s), title, publication source (PeerJ) and either DOI or URL of the article must be cited.
License URL: https://creativecommons.org/licenses/by/4.0/

Keywords: Disturbance, Diversity, Local, Regional, Rotenone, Richness, Ponds, Aquatic, Recovery, Restoration

Funding: National Science Foundation DEB-0816113 Tyson Research Center, and Washington University This work was supported by funding from the National Science Foundation (DEB-0816113 to KGS), Tyson Research Center, and Washington University. The funders had no role in study design, data collection and analysis, decision to publish, or preparation of the manuscript.

==============================
The effects of disturbance on local species diversity have been well documented, but less recognized is the possibility that disturbances can alter diversity at regional spatial scales. Since regional diversity can dictate which species are available for recolonization of degraded sites, the loss of diversity at regional scales may impede the recovery of biodiversity following a disturbance. To examine this we used a chemical disturbance of rotenone, a piscicide commonly used for fish removal in aquatic habitats, on small fishless freshwater ponds. We focused on the non-target effects of rotenone on aquatic invertebrates with the goal of assessing biodiversity loss and recovery at both local (within-pond) and regional (across ponds) spatial scales. We found that rotenone caused significant, large, but short-term losses of species at both local and regional spatial scales. Using a null model of random extinction, we determined that species were selectively removed from communities relative to what would be expected if species loss occurred randomly. Despite this selective loss of biodiversity, species diversity at both local and regional spatial scales recovered to reference levels one year after the addition of rotenone. The rapid recovery of local and regional diversity in this study was surprising considering the large loss of regional species diversity, however many aquatic invertebrates disperse readily or have resting stages that may persist through disturbances. We emphasize the importance of considering spatial scale when quantifying the impacts of a disturbance on an ecosystem, as well as considering how regional species loss can influence recovery from disturbance.

Introduction

Given the magnitude of anthropogenic activities on natural systems (Vitousek et al., 1997), understanding if and how biodiversity recovers from disturbances is an important focus of ecology and conservation biology. The loss and recovery of biodiversity in response to disturbances has been extensively studied in multiple systems (Niemi et al., 1990; Yount & Niemi, 1990; Lake, 2000; Chazdon, 2003; Dunn, 2004; Flinn & Vellend, 2005). However, the vast majority of these studies have focused strictly on the negative effects of disturbance on local, or alpha, species richness, that is, the average number of species that occur in a particular habitat or location. Less recognized is the possibility that short-term, widespread environmental disturbances can alter species diversity at regional scales, or that species composition can be altered (Hamer & Hill, 2000; Smith, Lips & Chase, 2009). Because regional diversity can dictate which species are available to recolonize degraded sites (Palmer, Ambrose & Poff, 1997; Zobel, Van der Maarel & Dupré, 1998), the loss of diversity at regional scales has important implications to restoration ecology and the recovery of biodiversity after disturbances.

When both spatial scales of biodiversity are considered, it becomes clear that the loss and recovery of biodiversity in response to widespread environmental disturbances can follow multiple trajectories (Fig. 1A). For example, disturbance-induced biodiversity loss, as measured by species richness, can be restricted entirely to the local spatial scale. This can occur if a disturbance acts stochastically, randomly removing widespread species from some locations, but never from all locations in which they occur. As a result, some forms of disturbance may have strictly local effects with essentially no loss of regional biodiversity, even if the disturbance itself is widespread and “regional” (Powell, Chase & Knight, 2013). Other types of disturbances can reduce local species richness through the selective removal of species due to species-specific responses to an environmental stressor (e.g., Chase, 2007; Smith, Lips & Chase, 2009; Chase et al., 2009). As a result of this selective filtration of particular species from all disturbed sites, species will also be lost from the larger regional scale, potentially to a greater degree than would be observed at the local scale (Fig. 1A and Smith, Lips & Chase, 2009; Chase et al., 2009). Disturbance-induced local species loss will also result in the loss of regional species richness whenever a species is removed from the only site in which it occurs. The overall effect of a disturbance on biodiversity can therefore vary depending on the spatial scale considered and whether species loss from local communities is selective or not. Ultimately, the outcome of these interactions will determine whether the loss of biodiversity caused by a disturbance is restricted to local scales or leads to species loss at regional scales as well.

Figure 1 Species loss and recovery at local and regional spatial scales.

(A) Disturbance can result in a loss of species from local communities relative to control communities (solid black line), but can also have differing effects on regional species richness depending on the type of disturbance. If a widespread disturbance removes species from local communities but not from every site where they occur, local species richness will be lost, but regional richness across all sites can remain unaffected (sloped grey dashed line). However, if some species are selectively removed from every local community in which they occur, then both local and regional species richness will be lost. In the most extreme case, local and regional species richness would be identical (flat grey dotted line). Most disturbances, including those acting randomly, will result in species loss somewhere between the two extremes (grey dot-dash line). (B) After a disturbance, local species richness can recover from its post-disturbance level (denoted by x) relative to control communities (solid black line), but species richness may or may not recover at the regional level due to a number of factors including dispersal limitation of extirpated species and priority effects or invasion resistance. In an extreme example, if the same set of species recolonize each site after a selective disturbance, local species richness can recover without any recovery of regional species richness (flat grey dotted line). It is also possible for regional species richness to only partially recover to pre disturbance levels (grey dot-dash line).

Whether disturbances cause biodiversity loss only at local, or at local and regional spatial scales has important implications for restoration. Following a disturbance, the recovery of species richness depends on the restoration of suitable environmental conditions, species interactions within the disturbed area, and an adequate supply of propagules for the re-establishment of species (Palmer, Ambrose & Poff, 1997). Extensive research has examined the recovery of species richness at local spatial scales following a disturbance (reviewed in Niemi et al. 1990; Yount & Niemi, 1990; Lake, 2000; Chazdon, 2003; Dunn, 2004; Flinn & Vellend, 2005), including identifying the importance of regional species pools for the recovery of local species richness (Vellend, 2003; Patrick & Swan, 2011; Sundermann, Stoll & Haase, 2011; Thrush et al., 2013). However, far fewer studies have investigated the recovery of species richness at both local and regional scales (Patrick & Swan, 2011). This is an important gap in the theory and practice of ecological restoration. For example, in the case of a selective disturbance that leads to the loss of regional richness in addition to local richness, local richness can recover without complete recovery of regional richness (Fig. 1B). Species differ in their dispersal ability (e.g., Conrad et al., 1999), so the recovery of some species may be impeded or even precluded by the fact that they have to come from outside the affected focal region in order for recovery to occur. Additionally, even if a species disperses to a habitat they may be unable to successfully colonize due to priority effects or invasion resistance (Shurin, 2000; Chase, 2003; Chase, 2007). Consequently, if local richness is used to determine if restoration has been successful or complete, some species may still be missing from the regional spatial scale.

In this study we examined if loss of species from the regional level impedes the recovery of disturbed local communities, or if local recovery can occur without recovery at the regional level (Fig. 1B). We applied a chemical disturbance of rotenone, a piscicide commonly used for fish removal in aquatic habitats, to small fishless freshwater ponds. Although rotenone is often used to restore the biodiversity of historically fishless ponds by removing fish, rotenone is also widely recognized to have non-target effects on aquatic invertebrates (Hamilton, 1941; Smith, 1941; Cushing Jr & Olive, 1957; Almquist, 1959; Lindgren, 1960; Binns, 1967; Cook Jr & Moore, 1969; Anderson, 1970; Meadows, 1973; Claffey & Costa, 1974; Chandler Jr & Marking, 1982; Dudgeon, 1990; Beal & Anderson, 1993; Mangum & Madrigal, 1999; Melaas et al., 2001). We focus on these non-target effects of rotenone on aquatic invertebrate biodiversity, with the specific goal of assessing biodiversity loss and recovery at both local (within-pond) and regional (across ponds in a treatment) scales. We specifically predict that rotenone will be a selective disturbance, owing to past studies showing that aquatic taxa differ in their sensitivity to rotenone (Hamilton, 1941; Cushing Jr & Olive, 1957; Lindgren, 1960; Claffey & Costa, 1974; Chandler Jr & Marking, 1982; Dudgeon, 1990; Mangum & Madrigal, 1999; Melaas et al., 2001). As a result of this selectivity, we further predict that the number of species lost from the regional spatial scale (i.e., across all ponds in a treatment) will be greater than the number of species lost from the local pond scale. Finally, we also predict that recovery of regional species richness will be slower than recovery of local species richness owing to this large predicted loss of species from across all ponds in the treatment (i.e., regional scale)

Materials & Methods

Ethics statement

Our research was conducted at Washington University’s Tyson Research Center and approved by the Washington University in St. Louis Institutional Animal Studies Committee (protocol #20130058). All field research was conducted in accordance with the Missouri Department of Conservation guidelines under MDC Wildlife Collector Permit #15246. A licensed applicator (License #C18196) carried out the Rotenone application in accordance to the Missouri Department of Agriculture Pesticide Use Act.

Study site and experimental setup

This research was conducted in constructed ponds at Washington University’s Tyson Research Center (TRC) near St. Louis, MO, USA in 2012. In 2008, twelve ponds 6 m in diameter and 1 m in depth were constructed throughout the research center nested within similar terrestrial habitats of oak-hickory mixed forest. Ponds were separated by 500 m –800 m from each other and any other existing aquatic habitats. Ponds existed within a broader network of aquatic habitats, however, including temporary and permanent ponds, ephemeral and seasonal streams, and major rivers, the occupants of which provide the species pool for the experimental ponds in this study. At their start, ponds were stocked with a mixture of zooplankton and phytoplankton from more than 10 ponds in the region. Ponds were also stocked with gastropods and emergent and submergent macrophytes. Other macroinvertebrates (e.g., dragonflies, beetles, true bugs) were allowed to colonize naturally from the local species pool. Throughout the spring and summer of 2008, a large number of egg masses, larvae, and breeding adults of eight common species of anurans (Hyla versicolor, Acris crepitans, Pseudacris crucifer, P. triseriata, Rana sphenocephala, R. clamitans, R. sylvatica, and Anaxyurus americanus) were introduced from existing ponds at TRC to establish anuran populations. In 2011, at the conclusion of the initial research study conducted in these ponds, Ambystoma maculatum had naturally colonized 8 of the 12 ponds (Burgett, 2011). In order to ensure that there were no lingering effects of the previous study, we conducted pre-surveys of macroinvertebrates and zooplankton. The two treatments in this study (control and rotenone application) were stratified across past treatments to ensure that legacies of this past study would not bias our results. Initial pre-treatment sampling supported this, as we found no differences in aquatic community diversity or structure between ponds in the two treatment categories at the beginning of the study (see ‘Results’).

Rotenone treatment

Rotenone is commonly used as a pesticide and piscicide in North America. As a piscicide, it has been used for fish removal as a restoration strategy for small lakes, ponds, and other wetlands since the 1930s. Rotenone is most effective during the summer when water temperatures are above 21°C, which is also when zooplankton and macroinvertebrate richness and abundance are typically greatest. The half-life of rotenone in ponds is less than a day at 23°C (Gilderhus, Allen & Dawson, 1986; Gilderhus, Dawson & Allen, 1988), and it is largely considered the most benign fish control chemical agent widely used in management applications.

Rotenone was applied to six of the twelve ponds with the remaining six ponds left as untreated controls. A licensed pesticide applicator mixed a liquid formulation of Rotenone (Noxfish, 5% by volume active ingredient) solution to a 3 mg/L concentration, which was applied to each pond with a pesticide sprayer. This concentration is consistent with fish management strategies used in North America (Tate et al., 2003; Finlayson et al., 2010; Wynne & Masser, 2010). After application, the concentration of Rotenone that invertebrates were exposed to was approximately 0.15 mg/L. We sampled pond macroinvertebrates and zooplankton before Rotenone application on June 26th, 2012. The Rotenone treatment was applied on June 28, 2012. The first round of post-disturbance biodiversity sampling occurred on July 9, 2012 followed by a second round of biodiversity sampling 6 weeks post disturbance on August 13, 2012. We sampled macroinvertebrates and zooplankton again the following year on July 8, 2013 to monitor recovery.

Zooplankton sampling and data collection

We sampled zooplankton from ponds using a 2 L plastic pitcher. Eight 2 L samples of pond water were taken from the edge and center of the pond at varying depths and locations. We combined these samples and filtered them through an 80 µm zooplankton net to concentrate the sample to 50 mL. This sampling method is comparable to other studies in similar pond ecosystems (Steiner, 2004; Burgett & Chase, 2015). We preserved and dyed zooplankton samples immediately after collection using a standard Acid Lugol’s iodine solution.

We processed zooplankton samples after each field season. Each concentrated zooplankton sample was shaken and 10 mL was extracted and put into a petri dish. We used this 10 mL subsample to identify zooplankton species presence and abundance in a pond using a compound microscope at 20x–100x magnification. Species were identified to the lowest taxonomic level possible using keys from (Pennak, 1989; Thorp & Covich, 2010; Haney et al., 2013).

Macroinvertebrate sampling and data collection

We performed macroinvertebrate sampling using standard methods similar to those described in Chase et al. (2009). We placed a plastic cylinder stovepipe sampler measuring 36 cm in diameter and 1m tall in three randomly selected areas of the pond. We then used a 15 cm wide aquarium net to exhaustively sample macroinvertebrates from each stovepipe location. We sampled each stovepipe until 5 consecutive empty sweeps of the net. Every individual was collected and stored in one bin for all three stovepipes. In order to account for rare and fast swimming species, 20 additional net sweeps were conducted throughout the pond at varying depths and locations using a 30 cm-wide, 1 mm mesh D-net. These samples were collected in a separate container. We preserved all macroinvertebrates from these samples in 70% ethanol for later identification. We used a dissecting microscope for species identification to the lowest possible taxonomic group according to the keys of Needham, Westfall Jr & May (2000); Merritt, Cummins & Berg (2008); Thorp & Covich (2010).

Analyses

We combined zooplankton and macroinvertebrate data for each pond and each sampling period into species-by-pond matrices and conducted all analyses using these combined invertebrate (macroinvertebrate and zooplankton) biodiversity data. We quantified species richness as the number of species per pond (i.e., local richness) and using the Chao1 index (Chao, 1984) to account for any potential sampling bias. The Chao1 index estimates the number of species potentially missed during sampling based on the frequency of rare species in a sample. Chao1 values were calculated for each pond using the program Past (Hammer, Harper & Ryan, 2001). All analyses were run using both raw species richness and the Chao1 index.

We used T-tests in R (R Core Team, 2013) to verify that there were no differences in the number of species per pond between treatments prior to the addition of rotenone. In order to determine if rotenone addition impacted local richness both within and among sampling dates, we used linear mixed effects modeling (lme function in the nlme package in R (Pinheiro et al., 2013)). We chose a linear mixed effects modeling approach because local species richness values were normally distributed, and the mixed effects model accounted for the non-independence of the repeated measures of species richness in ponds over time. We selected the model with the lowest Akaike Information Criterion (AIC). The fitted model included fixed effects for differences in species richness between rotenone treatments, between sampling dates, and the interaction between rotenone treatment and sampling date, as well as the random effect of pond identity. The ANOVA results for the full model are provided in the (Table S1). When there were significant differences between treatments or among sampling dates, pair-wise multiple comparisons were used. Since the main results of this study are the same using Chao1 estimates of species richness or actual measured richness, only the results for the actual measured species richness are presented.

For analyses of pond species composition we calculated among-pond dissimilarity using Jaccard’s and Bray-Curtis dissimilarities for each treatment at each sampling date using the program PRIMER (Clarke & Gorley, 2006). We then used PERMANOVA to ask if pond invertebrate composition differed between control and rotenone ponds (Anderson, 2001; Anderson, Gorley & Clark, 2008; Chase, Burgett & Biro, 2010). We used a repeated measures PERMANOVA design with the fixed effects of rotenone treatment, sampling date, and the interaction between rotenone treatment and sampling date, and the random effect of pond identity nested within treatment. Post-hoc pair-wise PERMANOVA tests were used to identity if there were differences between control and rotenone ponds at each sampling date. Since the main effects of rotenone on species composition were the same using both Jaccard’s and Bray-Curtis dissimilarities, only the PERMANOVA results for Jaccard’s dissimilarity are presented. Finally, we visually assessed differences in species composition between treatments using nonmetric multidimensional scaling (NMDS) plots based on Jaccard’s dissimilarity in PRIMER (Clarke & Gorley, 2006).

We also aimed to test our prediction that species losses associated with the application with rotenone would be selective. Using R programming language (R Core Team, 2013), we applied a null model for random species extinction using the conceptual framework proposed by Smith, Lips & Chase (2009). This model develops a prediction for the expected number of extinctions under a null scenario of random extinction, given the magnitude of a series of local disturbances (in this case, rotenone). We compared actual species loss at the treatment level to this null prediction, which allowed us to determine if rotenone addition selectively removed species from local ponds resulting in a greater loss of regional species richness than would occur if species were randomly removed from local ponds. The number of species lost from each treatment pond between the pre sampling date and one week following rotenone addition was calculated and used to develop a random probability of extinction in the model. For each model run, the identity of the species that were removed (i.e., a local extinction) in each of the six-rotenone treatment ponds was chosen at random. The total regional (six pond) species richness was calculated after each run and saved. This model was executed 10,000 times, and mean expected regional richness and upper and lower 95% range values were calculated. We then compared these simulated values to our actual observed regional species richness after rotenone addition in order to obtain a p-value.

Results

Local and regional species richness

Prior to the application of rotenone, there were no detectable differences in local or regional richness between the two treatment groups (Fig. 2, “Pre Rotenone”; local richness: NS, t = 0.41, df = 8, p = 0.55). On average, each individual pond had 17.4 invertebrate species (SD = 4.1), and the control treatment group of six ponds had a regional species richness of 49, while the rotenone treatment group had a regional richness of 50 species of invertebrates.

Figure 2 Local and regional species richness in rotenone and control ponds.

Average local (1 pond) and regional (6 ponds) invertebrate species richness over time. The Pre Rotenone sampling occurred two days prior to rotenone application. The other sampling dates are denoted by the time after rotenone application. Lines connecting data points are included to assist the reader and do not denote a linear relationship. Error bars denote standard error.

The addition of rotenone had a significant effect on local species richness (p < 0.008, F1,10 = 11.20, Table S1; Fig. 2), but this effect varied across sampling dates (rotenone × sampling date interaction, p < 0.03, F2,20 = 4.35, Table S1; Fig. 2). One week after the application of rotenone to the six treatment ponds, mean local species richness was 48% lower in rotenone treatment ponds than in control ponds (Figs. 2 and 3; t = 4.47, df = 10, p < 0.005). Rotenone treatment ponds had, on average, 9.5 species fewer than control ponds. After the rotenone treatment, regional species richness of treated ponds dropped by a large margin, almost 40%, having only 31 species across the six rotenone-treated ponds as compared to 50 species across the six control ponds (Fig. 2). These results are consistent with our hypothesis that a greater number of species would be lost from the regional than local scale following rotenone addition (Fig. 3).

Figure 3 Species loss at local and regional spatial scales.

The scaling of local and regional richness in control and rotenone ponds one week after rotenone addition and the null model expectation for species loss. Error bars represent standard error for average local species richness and the 95% range for the null model expectation of regional species richness.

Six weeks after the application of rotenone, mean species richness was still significantly lower in rotenone ponds than in control ponds (14.5 versus 18.8 species, respectively; t = 2.37, df = 10, p < 0.05, Fig. 2). However, paired t-tests revealed that local species richness of rotenone ponds was similar to species richness in both rotenone ponds prior to application of rotenone and rotenone ponds one week after application. Qualitatively, the recovery of species richness at the local scale was more moderate than at the regional scale, with increases of 4.5 and 13 species, respectively.

One year after the application of rotenone to half the ponds, mean invertebrate species richness was similar among control and treatment ponds and no longer statistically distinguishable (NS, t = 0.65, df = 10, p > 0.5, Fig. 2). Further, the regional species richness across the six rotenone ponds had increased to 54 species, exceeding the regional species richness of the control ponds at that date.

Species loss null model

The fact that the loss of regional richness was greater than the loss of average local richness suggests that rotenone selectively filters invertebrates from the affected ponds. This selectivity is further supported by comparison of actual regional species loss to our simulation of random species loss (Fig. 3). A random pattern of local extirpations would have resulted in a regional species richness of 40 species (mean: 39.61 95% CI [39.57–39.65]). However, significantly more species were lost from the six-rotenone treatment ponds than would be expected if local extirpations were nonselective (p < 0.0001).

Species composition

Selective extirpations among ponds would be expected to lead to differences in community composition between control and treatment ponds, which is what we observed (Fig. 4). Prior to the application of rotenone, ponds from both treatments were intermingled in ordination space and could not be distinguished (Fig. 4). We found a significant effect of sampling date (PERMANOVA, F2,20 = 2.95 and P = 0.0001) and an interaction between rotenone and sampling date (PERMANOVA, F2,20 = 1.56 and P = 0.0089) on invertebrate species composition. One week after rotenone application, control and rotenone ponds were significantly different in their invertebrate species composition as measured by Jaccard’s dissimilarity (Fig. 4; PERMANOVA, F1,10 = 1.80 and P = 0.01, Table S2). Control and rotenone ponds continued to diverge in their species composition six weeks after rotenone application (Fig. 4; PERMANOVA, F1,10 = 1.75 and P = 0.006, Table S2). One year after the rotenone treatment, invertebrate community composition was again indistinguishable between the two treatments, providing further evidence that re-colonization and recovery of species richness and pond community structure relative to control ponds had occurred.

Figure 4 Invertebrate composition of rotenone and control ponds.

NMDS plots of invertebrate species composition in ponds at each sampling date as measured by Jaccard’s dissimilarity. Differences between control (black circles) and rotenone (open circles) ponds was determined using PERMANOVA.

Discussion

Overall, we found that rotenone causes a significant short-term loss of aquatic species richness at local and regional scales, resulting in changes to aquatic community structure. Yet despite the large loss (ca. 38%) of regional species richness from rotenone-treated ponds, the recovery of species richness at local and regional scales was not hindered and recovery was relatively complete by all measures within one year after the rotenone disturbance. We discuss each of these main results below.

The application of rotenone reduced aquatic invertebrate species richness at local and regional spatial scales and significantly changed pond species composition. We expected this result, given rotenone’s known toxicity to non-target aquatic invertebrates. The loss of species richness and change in species composition of local ponds was similar to previous observational and experimental studies, which found negative effects of rotenone on non-target species (Hamilton, 1941; Smith, 1941; Cushing Jr & Olive, 1957; Almquist, 1959; Lindgren, 1960; Binns, 1967; Cook Jr & Moore, 1969; Anderson, 1970; Meadows, 1973; Claffey & Costa, 1974; Chandler Jr & Marking, 1982; Dudgeon, 1990; Beal & Anderson, 1993; Mangum & Madrigal, 1999; Melaas et al., 2001). Although rotenone reduced species richness at both spatial scales, there was a larger than expected number of species lost from the regional level than the local level due to the selective nature of the rotenone disturbance, as expected (Fig. 3). We found a reduction in pond occupancy for a few taxa with known lethal concentrations or 24-hour LC50 concentrations of rotenone at or below what was added in this experiment (Hamilton, 1941; Chandler Jr & Marking, 1982). Specifically, for example, the occupancy of Notonecta irrorata was reduced from four out of six ponds to only one pond after rotenone application, while species in the class Hirudinea (leeches) were reduced from three to zero ponds, and the odonate Pachydiplax longipennis was extirpated from the four ponds in which it occurred (Table S2). Although documented only rarely, this kind of selective removal of species from a focal region is not unique to this rotenone disturbance, as a similar pattern has been documented in other systems subjected to disturbances including drought (Chase, 2007), emerging disease (Smith, Lips & Chase, 2009), and flooding (Lepori & Malmqvist, 2009). A critical question in restoration and conservation ecology is whether selectively extirpated species can successfully recolonize from the species pool, post-disturbance, or whether they are precluded from recolonization. Mechanisms that could potentially lead to the failure of species to recolonize include their extirpation from the region and the regional species pool, dispersal limitation, or priority effects and invasion resistance within local communities.

We found that local and regional species richness began to recover substantially six weeks after the application of rotenone. However, there was a larger recovery of species richness at the regional level than the local level (Fig. 2). This difference in recovery between spatial scales suggests that multiple species colonized a small subset of ponds, allowing regional species richness to increase more rapidly than local species richness. This is further supported by the increase in divergence of species composition between control and rotenone ponds six weeks after rotenone application compared to one week after application.

One year after rotenone application, local richness, regional richness, and species composition of invertebrate communities had recovered and were indistinguishable from control ponds. The full recovery of local pond invertebrate communities one year after rotenone application was not surprising as local recovery has been documented in previous studies (Beal & Anderson, 1993; Melaas et al., 2001), and many of these species readily disperse (Bilton, Freeland & Okamura, 2001; Havel & Shurin, 2004; Louette & De Meester, 2005; Van De Meutter, De Meester & Stoks, 2007) or have resting stages which may persist through disturbances (Brendonck & De Meester, 2003). Although each of the ponds in this study was at least 500 m away from other lentic habitats, the study site was located within a broader network of aquatic habitats, which may have provided a source of colonists. Additionally, some multivoltine species are capable of internal recovery via reproduction following disturbances (Hanson et al., 2007; Trekels, Van De Meutter & Stoks, 2011), which could have also contributed to the recovery of species richness in these ponds. Rotenone degrades quickly (Gilderhus, Allen & Dawson, 1986; Gilderhus, Dawson & Allen, 1988) and our results suggest that this allows any legacy effects it had on pond communities to be quickly erased by colonization and internal recovery dynamics.

Very few studies have examined the recovery of regional species diversity after disturbances. We predicted that the loss of species from a set of 6 rotenone-treated ponds would lead to a slow recovery of regional species diversity in those ponds for at least two reasons. First, priority effects that preclude the recolonization of extirpated taxa may occur as a result of selective disturbances, leading to high similarity among disturbed habitats that persists as an alternative stable state (Chase, 2003; Chase, 2007). Although our rotenone-treated ponds were highly similar up to six weeks after the rotenone treatment (Fig. 4) and were consistent with the predication that rotenone treatment provides a selective disturbance (Fig. 3), this effect did not persist as an alternative stable state, suggesting that priority effects were not significant in our study.

Second, we predicted that at least some of the regionally-extirpated species from the rotenone-treated ponds would face some dispersal limitation, resulting in a delay in their recolonization, potentially precluding the recovery of pond richness and composition to pre-disturbance levels. Organisms in aquatic systems vary greatly in their dispersal abilities, with some volant actively-dispersing species and other taxa that disperse passively (Bilton, Freeland & Okamura, 2001; De Bie et al., 2012), and recovery is expected to take longer with more dispersal limited taxa. Habitat isolation is also known to have a negative impact on the recovery of species richness following a disturbance (Caquet et al., 2007; Trekels, Van De Meutter & Stoks, 2011), particularly for species that do not have the potential for internal recovery (Trekels, Van De Meutter & Stoks, 2011). Although a legacy of the rotenone treatment persisted for at least 6 weeks, it was not observed a year after the rotenone treatment, which suggests that dispersal limitation and pond isolation were not important factors in our study after one year of recovery.

Despite the rapid recovery of regional species richness and community composition that we documented in this study, a similarly rapid recovery of local and regional species richness would not necessarily be expected for all disturbance scenarios. Even if disturbed habitats are not isolated, widespread disturbances that affect the entire regional species pool (e.g., acid rain, climate change) may reduce the source pool of potential colonists (Keller & Yan, 1998; Gray & Arnott, 2011), which would further impede recovery, potentially leading to the permanent loss of some species from the regional scale In contrast, if species are lost only from some sites in the landscape (ponds in this case), but still persist in the species pool as potential colonists, as in the case of this study, then recolonization by these species may occur very quickly. Ultimately, the rate of recovery will depend on the interaction between the dispersal ability of extirpated species, isolation of the disturbed habitat, the suitability of local habitats for recolonization (including the presence of priority effects), and the presence of a source pool of potential colonists. Our results are relevant to the widespread use of rotenone as a piscicide for the restoration of ponds that were historically fishless but now contain fish. Rotenone has broad and well-documented negative effects on non-target aquatic organisms such as zooplankton and macroinvertebrates. Despite this, our results suggest that rotenone use for fish removal from aquatic habitats does not pose a long-term threat to aquatic biodiversity assuming that recolonization from other habitats in the landscape is possible. However, our results may not apply to extremely isolated habitats or to the universal application of rotenone across an entire landscape of aquatic habitats. In each of these cases, recolonization of locally extirpated species may be more restricted resulting in delayed recovery of diversity. Moreover, owing to well-documented effects introduced fish have on aquatic diversity and composition (Brooks & Dodson, 1965; Hall, Cooper & Werner, 1970; Crowder & Cooper, 1982; Knapp, Matthews & Sarnelle, 2001; Chase et al., 2009), the temporary loss of biodiversity seen in our study may not be observed in habitats that are already impacted by fish.

Despite our conclusion that local and regional diversity and community structure recovered quickly after the disturbance in our study, we emphasize the importance of considering the impacts of disturbance at different spatial scales, including a specific consideration of how regional species loss can influence recovery from disturbance. Future research disentangling the relative importance of these processes for the recovery of species richness following a disturbance would have important implications for restoration practices, especially in other systems where isolation and dispersal limitation may present more significant barriers to recolonization.

Supplemental Information

Table S1 Linear mixed effects model summary statistics for the effects of rotenone on pond species richness and the Chao1 estimate of diversity

The fitted model included the fixed effects of rotenone treatment, sampling date, and the interaction between rotenone treatment and sampling date, as well as the random effect of pond identity. P-values <0.05 are indicated in bold.

Click here for additional data file.

Table S2 Species occupancy in rotenone ponds before rotenone application and one week and one year after rotenone

Species can have a maximum occupancy of six ponds. The complete loss of a species from rotenone ponds is denoted by orange shading. The colonization or recolonization of a species is denoted by blue.

Click here for additional data file.

File S1 Raw data of species abundances in ponds over time

Species by site matrix of pond invertebrate abundances.

Click here for additional data file.

File S2 R code for extinction null model

Click here for additional data file.

We would like to thank Tyson Research Center and its staff for use of their facilities and equipment. We thank Jennifer Heemeyer for help with field and lab work, and Chloe Pinkner, Sarah Jacobs, Kelly Muething, Alyssa Welker and the Tyson Environmental Research Fellowship (TERF) Program for help with pond sampling. John Wingo advised with Rotenone application procedures.

Additional Information and Declarations

Competing Interests

Author Contributions

Animal Ethics

Field Study Permissions

Data Availability

The authors declare there are no competing interests.

Lauren M. Woods conceived and designed the experiments, analyzed the data, wrote the paper, prepared figures and/or tables, reviewed drafts of the paper.

Elizabeth G. Biro conceived and designed the experiments, performed the experiments, analyzed the data, wrote the paper, reviewed drafts of the paper.

Muxi Yang conceived and designed the experiments, performed the experiments, analyzed the data, wrote the paper.

Kevin G. Smith conceived and designed the experiments, performed the experiments, wrote the paper, reviewed drafts of the paper.

The following information was supplied relating to ethical approvals (i.e., approving body and any reference numbers):

Our research was approved by the the Washington University in St. Louis Institutional Animal Studies Committee (protocol #20130058).

The following information was supplied relating to field study approvals (i.e., approving body and any reference numbers):

All field research was conducted in accordance with the Missouri Department of Conservation guidelines under MDC Wildlife Collector Permit #15246.

The following information was supplied regarding data availability:

The raw data has been supplied as a Supplemental Dataset.

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
