# Peer review of "Does regional diversity recover after disturbance? A field experiment in constructed ponds"

_PeerJ, doi:10.7717/peerj.2455_

## Round 0.1 · original submission · Major Revisions

· Academic Editor

Major Revisions

The reviewers and I appreciate the work you have accomplished. There were, however, many suggested areas for revision and improvement. I find the comments by the reviewers to be reasonable, and I ask that you address their concerns wherever possible in revising the manuscript.

Reviewer 1 ·

Basic reporting

The introduction is ok, but requires expansion. Particularly, the article would benefit if explanations on how disturbance may affect diversity were provided.

The figures are relevant, but may need some adjustments. On figure 4, please indicate the species that may be driving the difference between the treatments the most. Figure 1 does not convey the message very clearly. Perhaps have a single plot with two or three lines (and no shaded triangle): one line with slope = 0 where local and regional species richness are the same; a second line where the there is a difference between local and regional richness. The change in local or regional species richness (either increasing or decreasing at each side) could be illustrated with the arrows.

The discussion can be shortened, particularly in the last three paragraphs, which tend to repeat themselves.

Additional comments are given on "general comments to authors".

Experimental design

The experiment was well designed and suitable statistical analyses were conducted.

Validity of the findings

The data seem ok, but I am not convinced about the interpretation of the results. The authors expected a slow recovery as if species would recolonize the treated ponds from outside the region. However, 6 other ponds nearby contained a similar species composition (as shown by their NMDS), and provided individuals for recolonization.

The authors also atempted to compare local vs regional species loss, and concluded that the a greater loss at the regional scale (rather than at the local scale) was surprising. This would only be surprising if all ponds contained the same species composition. As some species will not be present across all ponds, it is expected that the regional scale will have a greater number of species being lost than any individual pond.

Additional comments

This is an interesting paper that tests how disturbance affects local and regional species diversity. The authors have used a mesocosm approach where 6 ponds from a total of 12 had a strong pesticide applied. The authors compared invertebrate (benthic and zooplankton) species richness and composition between treatments and across time. They found that all ponds recovered after half year, and suggest that rotenone is suitable for restoration projects (when the project aims at eliminating the fish community to restore to original levels without fish), since species richness and composition showed clear sign of quick recovery. The authors further attempted to test if local species richness and composition (at the pond scale) would be more impacted than the regional scale (across all 12 ponds). I am not entirely convinced by how the authors have described their expectations. They mention surprise that invertebrates recolonized the treated ponds, but this should be expected given the existence of other invertebrate populations nearby. Given that the distances between ponds must have been standard, it is hard to say if further away ponds would have a slower recovery rate than nearby ponds.

Abstract
L20. Which effects of disturbance on diversity have been well documented?

Introduction
The authors mention that the effects of disturbance on diversity are well studied, and then mention the lack of studies at larger scales. While it may be true that disturbance-diversity relationships are well studied, it is not clear what the outcomes from those studies are. Disturbance has been shown to increase, decrease or even to have no effect on diversity due to a number of mechanisms that are often omitted from the hypothesis (see Fox 2013, TREE and reviews on the topic)

L 44. What are the conclusions/outcomes of these studies? Please, explain briefly what the link between disturbance and loss/recovery of species diversity is.

L54. It would be really important to state more clearly what your expectatio9ns regarding disturbance-diversity are, so that the illustration can be understood more easily. The example would then be easier understood in context of disturbance-diversity relationships if stated, for example, as “some disturbances reduce diversity, as measured by species richness, at the local scale only, when relatively widespread species are immune to the disturbance in other locations”.

L61. This sentence does not necessarily contradict the previous one. The main point is that some disturbances can occur that the local or at the regional level, therefore affecting species diversity at the two levels as well. Please, rephrase.

What if disturbance affects the local scale only, but ends up affecting the regional scale due to the role of that particular location as refuge, or storage of individuals that further colonize other locations?

L88-89. Here you are talking about diversity (species richness) and composition in the same sentence. The two may have different outcomes following a disturbance. An increase in species richness at either scale does not mean a recovery of the individual species that were lost.

L91. So, do you expect that alpha diversity will increase but not gamma diversity?

Methods

L188. How is Chao 1 calculated?

Results

L239. Standard deviation? Please, also provide the stats in the text for the comparisons between treatments.

L242. F-value?

L250-251. Move this sentence to discussion.

L254 and 260. I am confused with the results here. Maybe rephrase to: “Although mean species richness was still significantly lower in rotenone ponds than in control ponds (…), paired t-tests indicated that local species richness in rotenone ponds was similar to species richness in both rotenone ponds one week after rotenone addition and in rotenone ponds before rotenone application.”

Where is the difference described above coming from? Are the control ponds higher in species richness than before rotenone application? Has the variance in the treatments been reduced?

L258. “This suggests…” should move to discussion

L264. Please, provide a summary of the stats (F and p values) here.

L266. This sentence should be moved to the discussion.

L263. This sentence could be ended at “… between control and treatment ponds” after which you could report the F and p values.

L265. Statistically significant? Even if not, report the mean ± s.d.

L266. Discussion

Discussion

L289. I am not confident with the significance of this statement. In a collection of 12 ponds, if you remove species from half of it, you would expect that both individual ponds and all ponds together would lose species. This would not be the case only if all species were equally present in all ponds, in which case the removal of species from half the ponds would probably cause a change in cross-pond species distribution (some species become more abundant regionally than others), but not in total species richness. Thus, after applying a strong pesticide such as rotenone to half the ponds, it should not come as a surprise that the local and regional species richness would be affected negatively (unless, as said above, the species composition was the same across all 12 ponds). Maybe rephrase to indicate that you are aware of the implications in case species composition was the same across ponds?

L303. I am not sure I understand this argument. In my opinion, the reason the regional scale lost more species than the local scale is that different species were present in different ponds. As they were removed from the ponds, the regional scale summed more species being lost than each of the individual ponds.

L344-354. The argument here is that species would recolonize the ponds from outside the focal region, but this is not necessary if the same species were also present in the other non-treated ponds, as the NMDS seems to indicate.

L344. I think your expectations would have to be built on whether ponds differed substantially in species composition, thus leading to a slow recovery rate, or if they had similar species composition, leading to short recovery rate due to a storage effect.

Reviewer 2 ·

Basic reporting

This article claims “to test if loss of species from the regional species pool impedes the recovery of local populations, or if local recovery can occur without recovery at the regional level.” Although the article is very well written, the objective I have cited from the article shows one of my main issues with it: The whole article, including the research question(s), is written in a very general way. This starts with the introduction to the article, which refers to “biodiversity”, “multiple systems”, “environmental disturbance” etc. I´m not convinced at all that the performed experiment can provide answers to help understand local and regional biodiversity recovery after disturbances in general in any kind of ecosystem. Having said that, in my opinion most parts of the introduction should be rewritten, this time focusing on aquatic ecosystems only (mainly ponds), chemical (short-term, pulsed) disturbances (or at least discuss different types of disturbances and set the rotenone application into perspective to those, since I think it´s a very specific type of disturbance) and local/regional effects on aquatic biodiversity (mainly macroinvertebrates and zooplankton). Additionally, I would carefully reselect the citations: results from river studies for instance may not be applicable to pond ecosystems (but please check Jonathan Tonkin´s publications), or results from plant studies may not be applicable to macroinvertebrate/zooplankton recovery. Also, I recommend rephrasing the research question(s): The described experiment does not allow testing whether “local recovery can occur without recovery at the regional level”. To test this, recovery at the regional level would need to be restricted. Another main issue I have with the article is that the reader is left wondering about individual effects of macroinvertebrate and zooplankton depletion (and recovery). Although authors sampled both, analyses are performed on pooled data only, and taxa lists are missing altogether.
Unfortunately, I believe that figure 1 does not add any additional information to the article nor does it help conveying the concept of local and regional species richness. An interesting point is that the local and regional scale (on the x-axis) are connected (through the solid black line and dashed grey line), which means there must be some kind of spatial scale gradient among them. However authors miss to discuss this gradient. In figure 2, I miss an indication that shows when rotenone was added. And again in this figure, authors generously connect different measurements, although samples have been taken (quite) far apart from each other in time. Hence a linear relationship, at least between the 1 week, 6 weeks and 1 year sampling points, should not be assumed. The same holds for figure 3 (a gradient is indicated which I believe is not tested). In figure 4, descriptions of the axes are missing.
Regarding the Supporting Information: Not all files are cited in the article, and data descriptions in the individual files is missing altogether.

Experimental design

I believe that the research question, as it stands now, is too broad and should be refined (see above). Elaborating the knowledge gap more carefully and in more detail may considerably help in doing so. Authors may also include a figure depicting the experimental set-up. So far, very limited information on the spatial distribution of experimental units and natural habitats surrounding them (providing recolonisation potential) is given. However, such information is pivotal to convince the reader that the study includes local and regional aspects.
A short comment to the zooplankton sampling: I recommend including a reference here, since the sampling protocol seems not to be a standard one (using a plankton net for xy minutes for instance).

Validity of the findings

Regarding the performed analyses: From the description given, it is not clear whether a repeated measures ANOVA or a Generalized Linear Mixed Model was applied. The term “mixed model” is used in various ways in the literature, so this may just be a matter of better defining what has been done, which models have been tested, and which one fitted best (using AIC for instance). I also miss a description of the data exploration phase, e.g. testing normality etc. This will also help understanding whether an ANOVA can be applied or not. Same holds for the PERMANOVA approach: I´m missing the reasoning why this approach is the best one for the given data. Additionally, I recommend structuring the results section (and potentially the discussion section accordingly) in different parts adding a topic heading to each of them. This will make it much easier for the reader to follow the story.

---

## Round 0.2 · accepted · Accept

· Academic Editor

Accept

The authors appropriately addressed all of the reviewer's comments and I recommend this manuscript for publication. Well done!